# Trends in the Incidence and Mortality of Diabetes in China from 1990 to 2017: A Joinpoint and Age-Period-Cohort Analysis

**DOI:** 10.3390/ijerph16010158

**Published:** 2019-01-08

**Authors:** Xiaoxue Liu, Chuanhua Yu, Yongbo Wang, Yongyi Bi, Yu Liu, Zhi-Jiang Zhang

**Affiliations:** 1Department of Preventive Medicine, School of Health Sciences, Wuhan University, Wuhan 430071, China; liuxx019@163.com (X.L.); yuchua@whu.edu.cn (C.Y.); wangyb20172030@163.com (Y.W.); yongyib@aliyun.com (Y.B.); 2Department of Statistics and Management, School of Management, Wuhan Institute of Technology, Wuhan 430205, China; lyu429@163.com

**Keywords:** incidence, mortality, diabetes mellitus, joinpoint regression analysis, age-period-cohort effect, trends

## Abstract

**Background:** The prevalence of diabetes mellitus is rapidly increasing in China, but the secular trends in incidence and mortality remain unknown. This study aims to examine time trends from 1990 to 2017 and the net age, period, and cohort effects on diabetes incidence and mortality. **Methods:** Incidence and mortality rates of diabetes (1990–2017) were collected for each 5-year age group (from 5–9 to 80–84 age group) stratified by gender from the Global Burden of Disease 2017 Study. The average annual percentage changes in incidence and mortality were analyzed by joinpoint regression analysis; the net age, period, and cohort effects on the incidence and mortality were estimated by age-period-cohort analysis. **Results:** The joinpoint regression analysis showed that age-standardized incidence significantly rose by 0.92% (95% CI: 0.6%, 1.3%) in men and 0.69% in women (95% CI: 0.3%, 1.0%) from 1990 to 2017; age-standardized mortality rates rose by 0.78% (95% CI: 0.6%, 1.0%) in men and decreased by 0.12% (95% CI: −0.4%, 0.1%) in women. For age-specific rates, incidence increased in most age groups, with exception of 30–34, 60–64, 65–69 and 70–74 age groups in men and 25–29, 30–34, 35–39 and 70–74 age groups in women; mortality in men decreased in the younger age groups (from 20–24 to 45–49 age group) while increased in the older age groups (from 50–54 to 80–84 age group), and mortality in women decreased for all age groups with exception of the age group 75–79 and 80–84. The age effect on incidence showed no obvious changes with advancing age while mortality significantly increased with advancing age; period effect showed that both incidence and mortality increased with advancing time period while the period trend on incidence began to decrease since 2007; cohort effect on incidence and mortality decreased from earlier birth cohorts to more recent birth cohorts while incidence showed no material changes from 1982–1986 to 2012–2016 birth cohort. **Conclusions:** Mortality decreased in younger age groups but increased in older age groups. Incidence increased in most age groups. The net age or period effect showed an unfavorable trend while the net cohort effect presented a favorable trend. Aging likely drives a continued increase in the mortality of diabetes. Timely population-level interventions aiming for obesity prevention, healthy diet and regular physical activity should be conducted, especially for men and earlier birth cohorts at high risk of diabetes.

## 1. Introduction

Diabetes mellitus is a global health challenge in the 21st century [1,2,3]. As reported, global age-standardized prevalence of diabetes in men increased from 4.3% in 1980 to 9.0% in 2014 and in women increased from 5.0% to 7.9% [4]. In China alone, the age-standardized prevalence of diabetes increased to 5.04% for both men and women in 2017 [5]. Due to its large population, there were an estimated 3,338,131 new cases and 153,184 deaths of diabetes in China in 2017, accounting for 14.55% and 11.18% of all new cases and all deaths of diabetes worldwide, respectively [5,6]. To understand and control the burden of diabetes, it is necessary to analyzed the trends in the incidence and mortality of diabetes. China has undergone a rapid aging transmission, medical care and socioeconomic development. All these changes may impact the incidence or mortality rates differently for different age groups. Thus, we examined the trends in diabetes incidence and mortality rates for each 5-year age group stratified by gender from 1990 to 2017.

## 2. Materials and Methods

### 2.1. Data Source

The incidence and mortality rates of diabetes were obtained from the Global Burden of Diseases 2017 (GBD 2017) Study, which provided a comprehensive assessment of incidence, prevalence, and years lived with disability (YLDs) for 354 causes in 195 countries and territories from 1990 to 2017. Diabetes mellitus includes type 1 diabetes mellitus (T1DM) and type 2 diabetes mellitus (T2DM) in this study. The incidence and mortality of diabetes mellitus for all ages in different provinces was age-standardized by the GBD 2017 global age-standard population [5]. The original data, which GBD adapted to estimate incidence and mortality of diabetes mellitus in China, was mainly from the Cause of Death Reporting System of the Chinese Center for Disease Control and Prevention (CDC), Disease Surveillance Points (DSPs) and the Maternal and Child Surveillance System, which are considered to be nationally representative [7,8,9,10,11].

### 2.2. Joinpoint Regression Analysis

The identification of changes in time trend is an important issue in the analysis of mortality and incidence data, and such changes were described by joinpoint regression analysis. In this analysis, logarithmic transformation of the rates was carried out and the standard errors were calculated based on binomial approximation. To determine the magnitude of the time trends in incidence and mortality rates of diabetes, the average annual percent change (AAPC) and corresponding 95% confidence interval (CI) was evaluated using joinpoint regression analysis [12]. AAPC was calculated as a geometrically weighted average of various annual percent change (APC) values from the regression analysis [13]. This analysis was performed using ‘Joinpoint’ software from the Surveillance Research Program of the US National Cancer Institute.

### 2.3. Age-Period-Cohort (APC) Analysis

Incidence and mortality reflect not only the incidence and death risks experienced by the population in a current year but also the accumulation of health risks since birth. Common statistical analysis could not decompose these risks and health risks when estimating incidence and mortality [14,15]. APC analysis is developed to estimate net age, period, and cohort effects on incidence and mortality trends simultaneously [8,16]. APC model with intrinsic estimator (IE) method can be used to decompose three temporal trends and provides unbiased and relatively efficient estimation results [17,18]. In APC-IE model, the age-specific rates were appropriately recoded into successive 5-year age groups (5–9, 10–14, …, 80–84), consecutive 5-year periods from 1990 to 2017, and correspondingly consecutive 5-year birth cohort groups (1912–1916, 1917–1921, …, 2012–2016) to estimate net age, period, and cohort effects on incidence and mortality of diabetes. In this model, the groups under 5 years old for incidence were excluded, and groups under 20 years old and groups above 85 years old for mortality were excluded. The APC model can be expressed as:*Y_j_* = *µ* + *αage_j_* + *βperiod_j_* + *γcohort_j_* + *ε_i_*(1)
where *Y_j_* denotes the response variable—the net effect on incidence or mortality of diabetes for group *j*, *α*, *β* and *γ* denote the coefficients of age, period and cohort of the APC model, respectively, and *μ* denotes the intercept of the model. *ε_i_* denotes the residual of the APC model.

APC model was performed through the Stata 12.0 software (StataCorp, College Station, TX, USA). Deviance, Akaike Information Criterion (AIC) and Bayesian Information Criterion (BIC) were used to estimate the degree of model fitting.

## 3. Results

### 3.1. Descriptive Analysis of Incidence and Mortality Rates of Diabetes

Trends in the crude incidence rate (CIR), age-standardized incidence rate (ASIR), crude mortality rate (CMR) and age-standardized mortality rate (ASMR) in men and women at all ages for diabetes from 1990 to 2017 are depicted in Figure 1. The age-standardized incidence rates of diabetes increased from 1990 to 2006 and subsequently decreased from 2006 to 2017, and mortality experienced a slight increase in men and did not change materially in women during 1990–2017.

### 3.2. Trends in Age-Specific Incidence and Mortality Rates Using Joinpoint Regression Analysis

Table 1 shows the average annual percent change (APCC) in the incidence and mortality of diabetes for both men and women in China from 1990 to 2017. Age-standardized incidence rate rose by 0.92% (95% CI: 0.6%, 1.3%) in men and 0.69% (95% CI: 0.3%, 1.0%) in women, and age-standardized mortality rate rose by 0.78% (95% CI: 0.6%, 1.0%) in men and declined by 0.12% (95% CI: −0.4%, 0.1%) in women over the last decades. For age-specific rates, incidence increased in most age groups (from age group 10–14 to 80–84), with exception of 30–34, 60–64, 65–69 and 70–74 age groups in men, and 25–29, 30–34, 35–39 and 70–74 age groups in women; mortality in men decreased for the younger age groups (from 20–24 to 45–49 age group) and increased for the older age groups (from 50–54 to 80–84 age group). Mortality in women almost decreased for all age groups with exception of the age group 75–79 (AAPC = 0.73%, 95% CI: 0.4–1.0%) and 80–84 (AAPC = 2.34%, 95% CI: 2.0–2.7%) during the period.

### 3.3. The Age, Period, and Cohort Effects on Incidence and Mortality Using Age-Period-Cohort Analysis

The APC-IE analysis estimated coefficients for the age, period, and cohort effects (Table A1). These coefficients were then calculated to their exponential value (exp(coef.) = ecoef.) that denoted the incidence and mortality relative risk (RR) of a particular age, period, or birth cohort relative to each average level [19] (Table 2). Figure 2 was also plotted to reflect the age, period, and cohort effect based on Table 2.

#### 3.3.1. Age Effect

After controlling for period and cohort effects, the net age effect on diabetes showed that the RR of mortality continuously increased with advancing age (from 20 to 84 years) (Figure 2a, Table 2); the RR of incidence increased with advancing age in men for the younger age groups (from 10–14 to 20–24 age groups) while slightly decreased for the older age groups (from 20–24 to 80–84 age groups) (Figure 2a), and in women, the changes could be divided into two slight decreases and two slight increases. The net age effect on diabetes indicated that its mortality significantly increased with advancing age, and no material changes in its incidence. From 5–9 to 80–84 age group, the RR of incidence increased by 16.38 times and 11.18 times in men and women, respectively; from age group 20–24 to 80–84, the RR of mortality increased by 69.59 times and 45.98 times in men and women, respectively.

#### 3.3.2. Period Effect

The net period effect presented slight increasing incidence and mortality trends in men and women (Figure 2b, Table 2). During the period of observation, the RR of diabetes incidence increased by 1.29 and 1.36 times in men and women, respectively, which indicated that the incidence increased with advancing time period. The changes of period effect on the incidence could be divided into one accelerating increase from 1992 to 2007 and one slight decrease from 2007 to 2017. The RR of mortality increased by 1.64 and 1.15 times in men and women, respectively; for men, the mortality continuously increased with time period, while women showed two slight decreases and two slight increases during the period 1992–2017. Overall, the net period effect on diabetes showed the incidence and mortality of diabetes increased with advancing time period.

#### 3.3.3. Birth Cohort Effect

The cohort effect presented the incidence risk continuously decreased from 1912–1916 to 1982–1986 birth cohort and subsequently showed no material changes from 1982–1986 to 2012–2016 birth cohort in both men and women (Figure 2c, Table 2); the mortality risk decreased from 1927–1931 to 1997–2001 birth cohort in men, while increased from 1912–1916 to 1932–1936 birth cohort and subsequently decreased to the 1997–2001 birth cohort among women. From the earlier birth cohorts to more recent birth cohorts, the RR of diabetes incidence decreased by 47.76% and 63.22% in men and women, respectively; the RR of mortality decreased by 91.16% and 90.99% in men and women, respectively.

## 4. Discussion

The present study used longitudinal data from the Global Burden of Disease Study to investigate age, period, and cohort effects of diabetes mellitus incidence and mortality trends in China between 1990 to 2017. The present study found, that diabetes mortality decreased in younger age groups and increased in older age groups, while diabetes incidence increased in most age groups during the last decades in China. Through APC model, the net age effect showed no material changes with advancing age for the incidence of diabetes while the mortality increased with advancing age both in men and women; the net period effect showed both the incidence and mortality of diabetes increased with advancing time period but the period trend on the incidence began to decrease since 2007; the net cohort effect presented the incidence and mortality decreased from earlier birth cohorts to more recent birth cohorts while the incidence showed no material changes from 1982–1986 to 2012–2016 birth cohort. Therefore, the three trends on the incidence and mortality of diabetes were discussed preliminarily in the following section, which could provide epidemiology evidences for understand on reasons of increasing prevalence of diabetes.

### 4.1. Age Effect

Age effect on the mortality of diabetes showed it increased with advancing age for both men and women, China’s aging transition may intensify this situation [20], and Chinese rapid aging is observed from 1980 to 2010 in China [21]. Previous studies have reported that older people with high-risk diabetes mellitus mortality [22], frailty is associated with increased mortality as diabetes mellitus and frailty are two conditions that frequently occur concurrently and are increasingly prevalent in the older patients [23]. Moreover, diabetes complications and co-morbidities are more frequent in old diabetics compared to their young counterparts, and the most frequent are cardiovascular diseases [24,25] and the most bothersome are visual and cognitive impairments [26]. All these factors may impact the net age effect on mortality that substantially increased with advancing age (from 20–24 to 80–84 years). However, it’s worth noting that the age effect on incidence is not significant in both men and women.

### 4.2. Period Effect

Period effect is usually influenced by a complex set of historical events and environmental factors. Our study reported for the first time that the incidence and mortality of diabetes increased over the last decades. Our findings were compatible with the many studies reporting an increasing prevalence of diabetes in China over the past decades [27,28,29,30,31].

Risk factors of diabetes mainly included family history of diabetes mellitus, age, obesity, and physical inactivity [32,33,34,35]. Nutritional changes and sedentary lifestyles are reported to be the main causes of the epidemic of diabetes in China [28]. Obesity is a known characteristic closely related to diabetes [27,32,33,34,36]. In addition, China seems to be at highest risk of type 2 diabetes in Asian countries [3,37], despite there has also been significant improvement in screening [38,39], treatment [40,41,42] and/or prevention of diabetes [43]. The reason for the increasing period trend was possibly related to the increasing prevalence of obesity in China. The China Health and Nutrition Survey showed that 184 million (14.7%) of Chinese were overweight and another 31 million (2.6%) were obese in 2002, out of a total population of 1.3 billion [44], and the prevalence of overweight has tripled in men and doubled in women from 1989 to 1997 among pre-school children [45], and the prevalence of obesity has also increased over the last decades among Chinese children and adolescents [46,47]. According to GBD 2017 study, the ASMR for diabetes attributable to high body-mass index shows a significantly increasing trend from 1990 to 2017 in China, while it sharply increased in global [5]. The trend in ASMR for diabetes attributable to high body-mass index through 1990–2017 was also plotted in the present study (Figure A1). The prevalence of obesity was very likely to contribute to the rising period trends in incidence and mortality of diabetes in China over the last three decades.

Apart from the obesity, family history of diabetes mellitus, age, physical inactivity [32], and occupational chemical exposure [48], alcohol consumption may also be the risk factor for diabetes mellitus incidence and mortality [49]. Engler et al. [50] illustrated that excessive alcohol consumption not only negatively impacts diabetes self-care adherence but also affects the course of diabetes, and drinkers are likely to have poor treatment adherence, leading to increased morbidity and mortality. Cullmann et al.’s study showed total alcohol consumption and binge drinking increased the risk of pre-diabetes and type 2 diabetes in men, while low consumption decreased diabetes risk in women [51]. Therefore, alcohol consumption is likely to be one of the factors explaining the period trend of diabetes.

Diabetes has also become a public health problem in rural China. The levels of awareness, treatment, and control of diabetes were relatively low [52]; moreover, the awareness and treatment were positively associated with age and were high in adults with a family history of diabetes and those who exercise frequently, but low for cigarette smokers and alcohol drinkers in China [40]. Treatment improvements didn’t seem to reduce the period trends in the incidence and mortality of diabetes, which was probably related to the differential diagnosis and management in China [53].

Finally, we found that the incidence of diabetes decreased from 2007 to 2017 in both men and women. The possible reason is worth being studied. A recent study reported that incidence of diabetes in Hong Kong appeared to be stable and there have been slight decreases through 2006 to 2014. This trend was also observed in other developed western and Asian countries [54], and again in keeping with patterns seen in the UK, USA, Korea and Denmark [55,56,57,58], and with a small reduction in obesity in Hong Kong over the same period [59]. The majority (94%) of the population in Hong Kong is ethnically Chinese as mostly second- or third-generation immigrants are from the southern Chinese province of Guangdong [60], which shows possible future trends in the incidence of diabetes for mainland China. Moreover, the decline in the incidence of diabetes in Hong Kong also suggests improved health assessment. As Hong Kong has a mixed public-private healthcare economy, which is based on the British National Health Service model with 94.2% of the funding derived from government general revenue [61], all residents can use public health care services at highly subsidized rates, who are provided services by the Hospital Authority which provides the majority of inpatient care (90% total bed-days and 80% of admissions), and 50% of specialist outpatient care, despite the private sector providing 70% of first-contact outpatient services [61,62]. These changes could possibly explain the decline in diabetes incidence since 2007.

### 4.3. Cohort Effect

Cohort effect represents variations, across groups of individuals born in the same year or years. These variations may arise when each succeeding cohort carries the imprints of physical and social exposures from gestation to old age [63,64]. Cohort effect on the incidence and mortality of diabetes showed a decreasing trend from earlier birth cohorts to more recent birth cohorts while the incidence showed no material changes from 1982–1986 to 2012–2016 birth cohort. The probable reason was that more recent birth cohorts received good education and had a stronger awareness of health and disease prevention, compared with earlier birth cohorts [65]. In addition, more studies reported that chemical exposure was associated with the risk of diabetes mellitus [48,66,67,68,69]. With the rapid industrialization in China in the last decades, the birth cohorts from 1982–1986 to 2012–2016 might have experienced a higher risk of diabetes due to increasing exposures to chemicals. The more recent cohorts for the incidence of diabetes showed no decreasing trends, which was possibly attributable to their high exposure to chemicals during the last decades.

This study has limitations. APC analysis with IE method is as an ecological study. We were thus unable to make causal inference. We only tried to bring forth scientific hypotheses regarding the causality of these trends in the incidence and mortality of diabetes, based on the available data and existing literatures. In addition, most patients with diabetes die of complications. We do not have details of diabetic complications when examining the trends in mortality rate from 1990 to 2017.

## 5. Conclusions

This study shows the mortality of diabetes decreased in younger age groups while increased in older age groups, but the incidence increased in most age groups. The net age effect showed the mortality increased with advancing age while no material changes are observed for its incidence; the net period effect generally showed both the incidence and mortality increased with advancing time period; the net cohort effect presented them decreased from earlier birth cohorts to more recent cohorts, but not for some individuals in more recent cohorts for the incidence. Overall, aging likely drives a continued increase in the incidence and mortality of diabetes. Timely population-level interventions aiming for obesity prevention, healthy diet and regular physical activity should be conducted, especially for men and earlier birth cohorts at high risk of diabetes.

## Figures and Tables

**Figure 1 ijerph-16-00158-f001:**
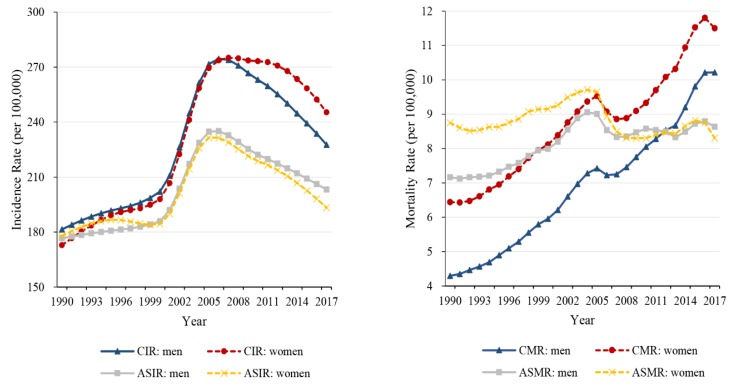
Trends in the crude rates and age-standardized rates for diabetes mellitus in men and women from 1990–2017, at all ages. CIR, crude incidence rate; ASIR, age-standardized incidence rate; CMR, crude mortality rate; ASMR, age-standardized mortality rate.

**Figure 2 ijerph-16-00158-f002:**
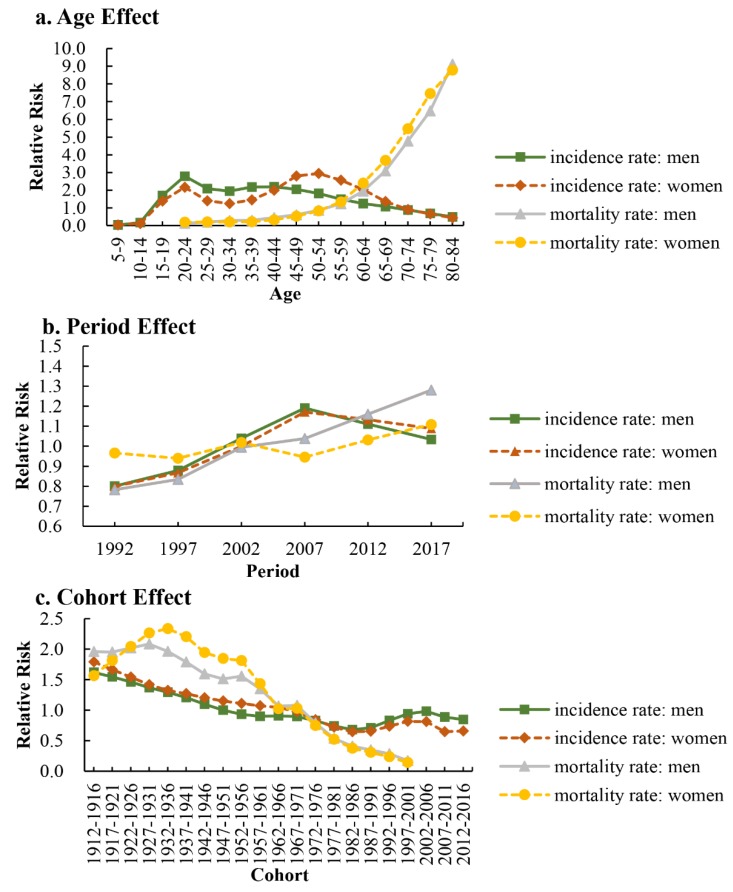
The incidence and mortality relative risks of diabetes due to (**a**) age; (**b**) period; and (**c**) cohort effects.

**Table 1 ijerph-16-00158-t001:** The average annual percent changes (AAPC) in incidence and mortality of diabetes, 1990–2017.

Age-Group (Year)	Incidence (95% CI)		Mortality (95% CI)	
	Men	Women	Men	Women
ASR	0.92 (0.6, 1.3)	0.69 (0.3, 1.0)	0.78 (0.6, 1.0)	−0.12 (−0.4, 0.1)
5–9	−0.63 (−0.8, −0.5)	−0.65 (−0.9, −0.4)		
10–14	0.05 (−0.4, 0.5)	0.63 (0.3, 0.9)		
15–19	3.27 (2.7, 3.9)	3.10 (2.5, 3.7)		
20–24	2.17 (1.8, 2.5)	1.79 (1.4, 2.1)	−3.98 (−4.7, −3.3)	−5.37 (−6.3, −4.5)
25–29	0.01 (−0.1, 0.1)	−0.75 (−0.9, −0.6)	−3.39 (−4.4, −2.4)	−4.88(−5.9, −3.8)
30–34	−0.02 (−0.3, 0.3)	−1.07 (−1.3, −0.8)	−3.45 (−4.3, −2.6)	−4.97 (−5.9, −4.1)
35–39	0.59 (−0.0, 1.2)	−0.76 (−1.1, −0.5)	−2.25 (−2.8, −1.7)	−3.77 (−4.4, −3.2)
40–44	0.92 (0.2, 1.6)	0.29 (−0.2, 0.7)	−2.42 (−2.8, −2.0)	−3.66 (−4.2, −3.1)
45–49	1.11 (0.5, 1.7)	1.25 (0.6, 1.9)	−0.52 (−0.9, −0.2)	−3.22 (−3.8, −2.6)
50–54	0.89 (0.3, 1.5)	1.26 (0.6, 1.9)	0.69 (0.3, 1.1)	−1.70 (−2.4, −1.0)
55–59	0.16 (−0.4, 0.8)	0.73 (0.1, 1.3)	0.21 (−0.3, 0.7)	−1.99 (−2.7, −1.2)
60–64	−0.36 (−0.9, 0.2)	0.36 (−0.1, 0.8)	0.80 (0.6, 1.0)	−1.36 (−1.6, −1.2)
65–69	−0.55 (−1.0, −0.1)	0.09 (−0.1, 0.3)	0.97 (0.6, 1.3)	−0.20 (−0.6, 0.2)
70–74	−0.41 (−0.8, −0.0)	−0.01 (−0.2, 0.2)	0.49 (0.2, 0.8)	−0.14 (−0.5, 0.2)
75–79	0.20 (−0.0, 0.4)	0.38 (0.1, 0.6)	1.41 (1.0, 1.8)	0.73 (0.4, 1.0)
80–84	0.55 (0.5, 0.6)	0.62 (0.3, 1.0)	1.83 (1.4, 2.3)	2.34 (2.0, 2.7)

CI, Confidence interval; ASR, age-standardized rate. Incidence and mortality for diabetes mellitus was age-standardized by the GBD 2017 global age-standard population.

**Table 2 ijerph-16-00158-t002:** The incidence and mortality relative risks of diabetes due to age, period, and cohort effects.

Factor	Incidence (RR 95% CI)		Mortality (RR 95% CI)	
	Men	Women	Men	Women
**Age**				
5–9	0.03 (0.02–0.04)	0.04 (0.03–0.06)		
10–14	0.16 (0.14–0.19)	0.11 (0.09–0.14)		
15–19	1.70 (1.57–1.83)	1.39 (1.27–1.52)		
20–24	2.79 (2.61–3.00)	2.16 (2.00–2.34)	0.13 (0.04–0.49)	0.19 (0.06–0.61)
25–29	2.09 (1.95–2.24)	1.39 (1.29–1.51)	0.19 (0.07–0.48)	0.20 (0.08–0.52)
30–34	1.94 (1.82–2.08)	1.24 (1.15–1.34)	0.28 (0.13–0.60)	0.21 (0.09–0.50)
35–39	2.18 (2.05–2.32)	1.45 (1.35–1.56)	0.29 (0.15–0.58)	0.21 (0.10–0.47)
40–44	2.20 (2.07–2.33)	2.00 (1.88–2.13)	0.45 (0.26–0.77	0.32 (0.17–0.59)
45–49	2.05 (1.94–2.17)	2.81 (2.66–2.96)	0.60 (0.39–0.94)	0.52 (0.33–0.84)
50–54	1.82 (1.72–1.92)	2.94 (2.80–3.09)	0.86 (0.60–1.24)	0.82 (0.57–1.20)
55–59	1.49 (1.41–1.58)	2.57 (2.45–2.70)	1.23 (0.91–1.66)	1.33 (1.00–1.78)
60–64	1.25 (1.18–1.32)	2.04 (1.94–2.14)	1.95 (1.53–2.47)	2.39 (1.91–3.00)
65–69	1.07 (1.01–1.14)	1.36 (1.29–1.44)	3.09 (2.51–3.80)	3.68 (2.99–4.53)
70–74	0.88 (0.83–0.94)	0.91 (0.85–0.97)	4.77 (3.86–5.90)	5.48 (4.36–6.88)
75–79	0.69 (0.64–0.74)	0.65 (0.61–0.70)	6.47 (5.06–8.28)	7.47 (5.66–9.85)
80–84	0.49 (0.45–0.53)	0.44 (0.40–0.48)	9.12 (6.76–12.31)	8.78 (6.24–12.37)
**Period**				
1992	0.80 (0.77–0.83)	0.80 (0.77–0.83)	0.78 (0.63–0.98)	0.97 (0.77–1.22)
1997	0.88 (0.85–0.91)	0.87 (0.84–0.90)	0.83 (0.71–0.98)	0.94 (0.80–1.10)
2002	1.04 (1.01–1.07)	1.00 (0.97–1.03)	0.99 (0.89–1.11)	1.02 (0.92–1.13)
2007	1.19 (1.16–1.23)	1.17 (1.14–1.20)	1.04 (0.93–1.16)	0.95 (0.85–1.05)
2012	1.11 (1.08–1.15)	1.13 (1.10–1.17)	1.16 (1.00–1.35)	1.03 (0.88–1.21)
2017	1.03 (1.00–1.07)	1.09 (1.05–1.13)	1.28 (1.04–1.58)	1.11 (0.89–1.39)
**Cohort**				
1912–1916	1.62 (1.35–1.95)	1.79 (1.48–2.17)	1.96 (1.24–3.09)	1.57 (0.97–2.53)
1917–1921	1.55 (1.36–1.76)	1.67 (1.46–1.90)	1.95 (1.34–2.85)	1.81 (1.22–2.68)
1922–1926	1.46 (1.32–1.62)	1.54 (1.39–1.71)	2.02 (1.46–2.80)	2.04 (1.46–2.85)
1927–1931	1.37 (1.26–1.49)	1.42 (1.30–1.55)	2.08 (1.56–2.79)	2.26 (1.68–3.04)
1932–1936	1.30 (1.20–1.40)	1.33 (1.23–1.43)	1.97 (1.49–2.60)	2.34 (1.77–3.09)
1937–1941	1.21 (1.12–1.30)	1.27 (1.19–1.36)	1.79 (1.35–2.38)	2.20 (1.65–2.94)
1942–1946	1.10 (1.02–1.18)	1.20 (1.13–1.29)	1.59 (1.15–2.20)	1.94 (1.40–2.70)
1947–1951	1.00 (0.93–1.08)	1.15 (1.08–1.23)	1.51 (1.04–2.20)	1.85 (1.25–2.72)
1952–1956	0.93 (0.86–1.00)	1.11 (1.03–1.19)	1.56 (1.01–2.41)	1.81 (1.15–2.86)
1957–1961	0.90 (0.83–0.97)	1.07 (1.00–1.15)	1.35 (0.81–2.25)	1.43 (0.83–2.48)
1962–1966	0.91 (0.84–0.98)	1.05 (0.97–1.13)	1.07 (0.59–1.94)	1.03 (0.53–1.99)
1967–1971	0.89 (0.83–0.97)	0.97 (0.90–1.06)	1.08 (0.56–2.09)	1.03 (0.49–2.15)
1972–1976	0.83 (0.76–0.90)	0.84 (0.77–0.92)	0.79 (0.37–1.69)	0.75 (0.32–1.75)
1977–1981	0.74 (0.68–0.80)	0.72 (0.66–0.79)	0.54 (0.21–1.41)	0.52 (0.17–1.58)
1982–1986	0.68 (0.62–0.75)	0.65 (0.58–0.72)	0.41 (0.12–1.39)	0.37 (0.09–1.56)
1987–1991	0.71 (0.64–0.78)	0.66 (0.59–0.73)	0.35 (0.08–1.56)	0.31 (0.05–1.77)
1992–1996	0.83 (0.75–0.92)	0.74 (0.66–0.83)	0.28 (0.03–2.34)	0.24 (0.02–2.35)
1997–2001	0.94 (0.84–1.05)	0.82 (0.72–0.92)	0.17 (0.00–16.78)	0.14 (0.00–14.53)
2002–2006	0.98 (0.86–1.12)	0.81 (0.69–0.95)		
2007–2011	0.89 (0.64–1.24)	0.65 (0.42–1.00)		
2012–2016	0.85 (0.33–2.18)	0.66 (0.26–1.68)		
Deviance	73.51 8.55 −182.09	105.89 8.87 −149.71	1.84 4.74 −189.85	3.04 4.74 −188.66
AIC				
BIC				

RR: Relative risk [RR = exp(coefficient)]; CI: Confidence interval; AIC: Akaike Information Criterions; BIC: Bayesian Information Criterions.

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
