# Peer review of "Trends in the Incidence and Mortality of Diabetes in China from 1990 to 2017: A Joinpoint and Age-Period-Cohort Analysis"

_ijerph, 2019, doi:10.3390/ijerph16010158_

Round 1
Reviewer 1 Report
General: The authors have identified an interesting research question. “Trends in incidence and mortality of diabetes mellitus in China from 1990 to 2017: a joinpoint and age-period-cohort analysis” is an interesting topic. The methods used are appropriate and the presentation of the data is well performed. Minor corrections and grammar needs to be improved before the article can be accepted.
Title is appropriate.
Abstract:
Authors mention that “incidence and mortality trends remain unknown”. In this recently published article “Diabetes in China: Epidemiology and Genetic Risk Factors and Their Clinical Utility in Personalized Medication authors mention about the various sources of incidence of diabetes.
Authors also mention about incidence and mortality rates in china in introduction section.
So this sentence needs to be corrected appropriately as it is contradicting what the authors are saying 2 different things.
Conclusion section of abstract authors mention “: Mortality decreases in younger age groups but increases in older age groups” , what was the comorbidities of the older age groups ? as older people have more comorbidities and higher mortality rates when compared to younger age groups
Introduction:
3. It will be prudent to also include the prevalence and mortality rates of CDI in Greece rather than using the statistics from USA. Or you can include both.
4. 3rd para – “In Greece we are lacking data on the prevalence of CDI in patients with IBD.” Is it only prevalence of CDI in IBD or prevalence of CDI. If the sentence mentioned above is true then please include the data for prevalence of mortality rates of CDI in Greece. If you do not have enough data then will need to modify the above sentence.
5. 3rd para – Change the last sentence from “In Greece we are lacking data on” to “In Greece, we lack data on…”
Materials and Methods:
Authors mention GBD 2017. What is GBD?
In age effect paragraph line 135-136 authors mention “These findings indicated that diabetes mellitus mortality significantly increased with 136 advancing age, and no material changes in its incidence.”
Diabetes mellitus by itself does not cause mortality as the complications of diabetes causes most of the mortality. Authors do not mention any of those.
Discussion:
When authors mention diabetes do they include all diabetics or DM type 1 or type or gestational DM. This needs to be mentioned clearly.
All in all, the analysis in this article is performed accurately and rigorously. Grammar needs to be rechecked throughout the manuscript.
Author Response
Dear Editor,
We would like to resubmit the revised manuscript entitled “Trends in incidence and mortality of diabetes mellitus in China from 1990 to 2017: a joinpoint and age-period-cohort analysis” (ID: ijerph-416123). Thank you for your letter and for the reviewers’ comments concerning our manuscript. Those comments are all valuable and very helpful for revising and improving our paper, as well as the important guiding significance to our researches. We have studied comments carefully and have made correction which we hope meet with approval. Revised portion are also clearly highlighted in the paper using “Track Changes”. The main corrections in the paper and the responds to the reviewer’s comments are as flowing:
Point 1:
Abstract:
Authors mention that “incidence and mortality trends remain unknown”. In this recently published article “Diabetes in China: Epidemiology and Genetic Risk Factors and Their Clinical Utility in Personalized Medication authors mention about the various sources of incidence of diabetes.
Authors also mention about incidence and mortality rates in china in introduction section.
So this sentence needs to be corrected appropriately as it is contradicting what the authors are saying 2 different things.
Response 1: We corrected “incidence and mortality trends remain unknown” as “secular trends in incidence and mortality remain unknown” in the revision. We downloaded the article that the reviewer recommended. In the article “Diabetes in China: Epidemiology and Genetic Risk Factors and Their Clinical Utility in Personalized Medication”, the authors did not report the incidence or mortality of diabetes. What they described is the prevalence of diabetes.
Point 2:
Conclusion section of abstract authors mention “: Mortality decreases in younger age groups but increases in older age groups”, what was the comorbidities of the older age groups? as older people have more comorbidities and higher mortality rates when compared to younger age groups
Response 2: We agree with the reviewer that older people have more comorbidities. We do not have the information. The causes of death, however, should be attributable to diabetes, according to the international guideline to death registry.
Point 3:
Introduction:
It will be prudent to also include the prevalence and mortality rates of CDI in Greece rather than using the statistics from USA. Or you can include both.
Response 3: We didn’t include the prevalence and mortality rates of CDI in Greece. We chose using the statistics from USA because the incidence and mortality rate of diabetes mellitus obtained from global burden of Diseases (GBD) 2017 provided a comprehensive estimation of 354 causes for 195 countries from 1990 to 2017, and incidence or mortality of diabetes mellitus for all ages (<1,>95) in different provinces was age-standardized by the GBD 2017 global age-standard population, and we could estimate the data based on age-standardized rates or age-specific rates of different populations by statistical methods comprehensively. The original data estimated by GBD for incidence and mortality of diabetes mellitus in China was mainly from the Cause of Death Reporting System of the Chinese Center for Disease Control and Prevention (CDC), Disease Surveillance Points (DSPs) and the Maternal and Child Surveillance System, which are considered to be nationally representative. While we could not estimate incidence or mortality of diabetes in China for the prevalence and mortality rates of CDI in Greece.
Point 4:
3rd para – “In Greece we are lacking data on the prevalence of CDI in patients with IBD.” Is it only prevalence of CDI in IBD or prevalence of CDI. If the sentence mentioned above is true then please include the data for prevalence of mortality rates of CDI in Greece. If you do not have enough data then will need to modify the above sentence.
Response 4: We didn’t draft this sentence in the 3rd paragraph in the paper.
Point 5:
3rd para – Change the last sentence from “In Greece we are lacking data on” to “In Greece, we lack data on…
Response 5: We didn’t draft this sentence in the 3rd paragraph in the paper.
Point 6:
Materials and Methods:
Authors mention GBD 2017. What is GBD?
In age effect paragraph line 135-136 authors mention “These findings indicated that diabetes mellitus mortality significantly increased with advancing age, and no material changes in its incidence.”
Diabetes mellitus by itself does not cause mortality as the complications of diabetes causes most of the mortality. Authors do not mention any of those.
Response 6: The Global Burden of Diseases Study 2017 (GBD 2017) includes a comprehensive assessment of incidence and mortality for 354 causes in 195 countries and territories from 1990 to 2017, and incidence or mortality of diabetes mellitus for all ages (<1,>95) in different provinces was also age-standardized by the GBD 2017 global age-standard population. The study conducts annual updates to incorporate new causes and data (including published literature, surveillance data, survey data, hospital and clinical data, and other types of data) and to improve demographic and statistical methods. Original data, which GBD adapted to estimate mortality and incidence of diabetes in China, was mainly from the Cause of Death Reporting System of the Chinese Center for Disease Control and Prevention (CDC), Disease Surveillance Points (DSPs) and the Maternal and Child Surveillance System. Data from GBD 2017 are considered to be nationally representative worldwide.
Line 135-136: We deleted “These findings indicated that diabetes mellitus mortality significantly increased with advancing age, and no material changes in its incidence.” The new version is “Generally, the complications of diabetes cause most of the mortality of diabetes patients. The net age effect on diabetes indicated that its mortality significantly increased with advancing age, and no material changes in its incidence.” In our study, we estimated the net age effect on diabetes mellitus incidence and mortality, and showed the relative risk of incidence and mortality. Our findings presented that the RR of diabetes mellitus mortality significantly increased with advancing age, and no material changes in the RR of diabetes incidence, which could indicate the changes of trends in diabetes incidence and mortality.
We agree with the reviewer that diabetes itself does not cause mortality. We added this to the Discussion.
Point 7:
Discussion:
When authors mention diabetes do they include all diabetics or DM type 1 or type or gestational DM. This needs to be mentioned clearly.
Response 7: We added the “Diabetes mellitus includes type 1 diabetes mellitus (T1DM) and type 2 diabetes mellitus (T2DM) in this study.” in the Material and Methods section.
Reviewer 2 Report
General comments:
In their secondary data analyses: “Trends in incidence and mortality of diabetes mellitus in China from 1990 to 2017: a joinpoint and age-period-cohort analysis”, Liu X. et al. used from the well-established Global Burden of Disease Study to investigate age, period and cohort effects of diabetes mellitus incidence and mortality trends in China between 1990 to 2017. The manuscript is well written, the methodology appears sound and the conclusions are valid. Therefore, I have only minor comments.
Detailed comments
Discussion: suggest to rewrite first sentence “This present study used longitudinal data from the Global Burden of Disease Study to investigate age, period and cohort effects of diabetes mellitus incidence and mortality trends in China between 1990 to 2017.”
Discussion: suggest to rewrite: “The present study found, that diabetes mortality decreased in younger age groups and increased in older age groups, while diabetes incidence increased in most age groups during the last decades in China.”
Discussion: given the fact, that nutritional changes and increasingly sedentary lifestyles are among the main causes of the diabetes epidemic in China, I would suggest to focus the last sentence of the conclusions accordingly: “Timely population-level interventions aiming for obesity prevention, healthy diet and regular physical activity should be conducted, especially for men and earlier birth cohorts at high risk of diabetes.”
Author Response
Dear Editor,
We would like to resubmit the revised manuscript entitled “Trends in incidence and mortality of diabetes mellitus in China from 1990 to 2017: a joinpoint and age-period-cohort analysis” (ID: ijerph-416123). Thank you for your letter and for the reviewers’ comments concerning our manuscript. Those comments are all valuable and very helpful for revising and improving our paper, as well as the important guiding significance to our researches. We have studied comments carefully and have made correction which we hope meet with approval. Revised portion are also clearly highlighted in the paper using “Track Changes”. The main corrections in the paper and the responds to the reviewer’s comments are as flowing:
Point 1:
Discussion: suggest to rewrite first sentence “This present study used longitudinal data from the Global Burden of Disease Study to investigate age, period and cohort effects of diabetes mellitus incidence and mortality trends in China between 1990 to 2017.”
Response 1: We revised the first sentence as“This present study used longitudinal data from the Global Burden of Disease Study to investigate age, period and cohort effects of diabetes mellitus incidence and mortality trends in China between 1990 to 2017”.
Point 2:
Discussion: suggest to rewrite: “The present study found, that diabetes mortality decreased in younger age groups and increased in older age groups, while diabetes incidence increased in most age groups during the last decades in China.”
Response 2: We revised the second sentence as“The present study found, that diabetes mortality decreased in younger age groups and increased in older age groups, while diabetes incidence increased in most age groups during the last decades in China”.
Point 3:
Discussion: given the fact, that nutritional changes and increasingly sedentary lifestyles are among the main causes of the diabetes epidemic in China, I would suggest to focus the last sentence of the conclusions accordingly: “Timely population-level interventions aiming for obesity prevention, healthy diet and regular physical activity should be conducted, especially for men and earlier birth cohorts at high risk of diabetes.”
Response 3: We agree with the reviewer. We rewrote it as “Timely population-level interventions aiming for obesity prevention, healthy diet and regular physical activity should be conducted, especially for men and earlier birth cohorts at high risk of diabetes”.